# Improvement in the Identification Technology for Asian Spongy Moth, *Lymantria dispar* Linnaeus, 1758 (Lepidoptera: Erebidae) Based on SS-COI

**DOI:** 10.3390/insects14010094

**Published:** 2023-01-16

**Authors:** Wenzhuai Ji, Fengrui Dou, Chunhua Zhang, Yuqian Xiao, Wenqi Yin, Jinyong Yu, D. K. Kurenshchikov, Xiue Zhu, Juan Shi

**Affiliations:** 1Beijing Key Laboratory for Forest Pest Control and Sino-French Joint Laboratory for Invasive Forest Pests in Eurasia, College of Forestry, Beijing Forestry University, Beijing 100107, China; 2Agricultural Integrated Service Centre, Agriculture and Rural Affairs Bureau, Fugong 673400, China; 3Guizhou Academy of Forestry, Guiyang 550005, China; 4Institute for Aquatic and Ecological Problems, Far East Brunch of Russian Academy of Science, 680000 Khabarovsk, Russia

**Keywords:** biological invasion, Asian spongy moth, rapid detection, quarantine, mitochondrial DNA

## Abstract

**Simple Summary:**

Preventing invasion of the Asian spongy moth (ASM) is a major priority for quarantine agencies in North America and elsewhere due to the significant damage and invasiveness of ASMs. ASMs are native to China and strict quarantine procedures have been implemented to prevent unintentional introductions of ASMs to other countries. As there are many closely related species in *Lymantria* (Hübner, 1819) that are morphologically similar to ASM, especially at immature stages, it is not always possible to determine accurately with morphology. Molecular techniques for rapid detection are, therefore, becoming an immediate need. The main procedures that have been used to identify ASMs are based on samples that have been under artificial culture for dozens of generations, and the artificial culture of these samples might have resulted in the appearance of genetic features that differ from those of natural populations. Moreover, sampling has only been conducted in a few locations in China; given that high genetic variation has been detected in ASMs from different regions in China in recent years, the current methods used for identifying ASMs are not sufficiently robust. In this study, we provide an updated method based on cytochrome oxidase I to enhance the efficacy of ASM identification. This improved approach permits identifications of ASMs to be made in 2–3 h using as little as 30 pg of genomic DNA. This method could, thus, be used to monitor the spread of ASM in real time and mitigate identification errors.

**Abstract:**

*Lymantria dispar* (Linnaeus, 1758), which is commonly known as spongy moth, with two subspecies, is found in Asia: *Lymantria dispar asiatica* and *Lymantria dispar japonica*, collectively referred to as the Asian spongy moth (ASM). The subspecies *Lymantria dispar dispar* occurs in Europe and is commonly known as the European spongy moth (ESM). The ASM is on the quarantine list of many countries because it induces greater economic losses than the ESM. Accurate identification is essential to prevent the invasion of ASM into new areas. Although several techniques for identifying ASMs have been developed, the recent discovery of complex patterns of genetic variation among ASMs in China as well as new subspecies in some areas has necessitated the development of new, improved identification techniques, as previously developed techniques are unable to accurately identify ASMs from all regions in China. Here, we demonstrate the efficacy of an improved technique for the identification of the ASM using ASM-specific primers, which were designed based on cytochrome oxidase I sequences from samples obtained from all sites where ASMs have been documented to occur in China. We show that these primers are effective for identifying a single ASM at all life stages and from all ASM populations in China, and the minimum detectable concentration of genomic DNA was 30 pg. The inclusion of other *Lymantria* samples in our analysis confirmed the high specificity of the primers. Our improved technique allows the spread of ASMs to be monitored in real time and will help mitigate the spread of ASMs to other areas.

## 1. Introduction

The accidental spread of human-mediated pests has been a major cause of the loss of forest resources. With an increase in global trade, a gradual increase in the number of invasive insects and the extent of their impact can be seen. Invasions of insects can be difficult to prevent given their small size and ability to hide in inaccessible places that are difficult to notice, such as agricultural products, plants and seeds transported as goods [1,2]. In addition, pests may hide in transport vehicles (e.g., ships, trains, planes) as stowaways, which facilitates covert spread along transport routes [3,4]. Generally, local biodiversity and the global economy are affected to a much greater degree by invasive species than native species [5,6,7,8,9]. Consequently, various stringent measures have been implemented in several countries to mitigate the spread and introduction of invasive species [2,10].

*Lymantria dispar* Linnaeus 1758 (Lepidoptera: Erebidae: Lymantriinae), which is commonly known as spongy moth, is an omnivorous pest, native to Europe, Asia, and North Africa, that has also been introduced to North America [11]. Spongy moths have been divided into three subspecies according to their distribution and whether the female is capable of flight. *Lymantria dispar dispar* Linnaeus 1758 mainly occurs in Europe and North America and is commonly referred to as European spongy moth (ESM). *Lymantria dispar asiatica* Vnukovskij 1926 and *Lymantria dispar japonica* Motschulsky 1860 are distributed in Asia and both are referred to as Asian spongy moth (ASM) [12]. Spongy moths are among “the world’s 100 worst invasive species” according to the International Union for Conservation of Nature [13,14]. The only forest pest under quarantine regulation by the USDA is spongy moth [15]. Masses of 200 to 1200 eggs are laid by females on the branches and trunks of trees and they have sometimes been observed to lay their eggs in shelters or houses [16,17]. Invasions of spongy moths can result in the complete defoliation of trees and bushes, which makes them more vulnerable to infection. In some cases, they can devastate large tracts of timber [18,19].

Since the mid-19th century when it was first introduced to North America, the European spongy moth (ESM) has induced substantial damage to forests [20,21]. The ESM has become one of the most economically significant forest pests on the east coast of the United States. Much money and time are invested by the U.S. government annually in ESMs control efforts [19]. The ASM has several biological features that make it more invasive than the ESM. For example, female ASMs are capable of sustained ascending flights of several kilometers, but no female European spongy moths (ESMs) have been found to be able to fly. ASM females show stronger phototaxis than ESM and this increases their likelihood of being attracted by port lights and laying eggs on docked ships. Furthermore, ASM larvae can feed on more than 500 host plants [22,23]. The effects of temperature on hatching vary among spongy moth subspecies. Eggs of the Asian spongy moth strain required less exposure to low temperatures to hatch compared to eggs of the ESM [24,25]. This ability to terminate dormancy in a relatively short period of time gives ASM the potential to be able to survive in a variety of climatic conditions.

Plant-conservation organizations in North America have made preventing accidental introductions of ASMs a priority [12,26]. Since 1991, the North American Plant Protection Organization and other countries (such as Chile and New Zealand) have imposed strict quarantines on those vessels passing through the ASM range at specific times [19,27]. As there is a general sense that spongy moths in China are considered *Lymantria dispar asiatica*, Chinese port authorities have specified thorough and strict measures to reduce ASMs in ports and to identify suspected moths and further prevent their spread. We have been working on a comprehensive study of the biology of and genetic variation in spongy moths as an economically important pest [24,28,29,30,31,32,33,34,35]. In recent years, we have found that spongy moths in China are not traditionally known only as *Lymantria dispar asiatica*, and there are also potential, surprising subspecies differentiation and genetic variation in China. The spongy moth of Southern China has evolved into a separate clade from the three subspecies and is distinguished from the *Lymantria dispar asiatica* by its wing surface pattern and male external genitalia [36] (unpublished data). Supported by multiple lines of evidence, including mitochondrial genes, nuclear genes and morphology, we speculate that the spongy moth of Southern China may be a new subspecies. In addition, populations in Yunnan, China, are more complex, with the undersides of their wings and male external genitalia distinctly different from those in other regions. In addition, we found that some populations in north-western China (Xinjiang, Kuduer) have a mixed pedigree of ESM and ASM. We speculate that the introduction of the ESM to Russia has resulted in genetic exchange with native ASMs and the production of hybrids. Some of our views have been confirmed by other scholars [37].

Several morphologically similar species in the genus *Lymantria* (Lepidoptera: Erebidae) in China can be confused with spongy moths, such as *Lymantria apicebrunnea* Gaede 1932, *Lymantria similis* Moore 1879, *Lymantria fumida* Butler 1877, *Lymantria xylina* Swinhoe 1903 and *Lymantria mathura* Linnaeus 1758. The highly similar morphology of egg masses and larvae of *Lymantria* increases the difficulty in identifying spongy moth. In some cases, ASM can be misidentified as other non-quarantine insects, such as *Lymantria apicebrunnea*. *L. apicebrunnea* has only been documented to occur in China according to previous studies [12]. Because few records of this species have been published, it did not receive much attention until 2019 when an outbreak was observed in parts of Yunnan Province, China. Initially, this species was misidentified as the ASM by control agencies. *L. apicebrunnea* larvae feed on walnuts, cherries, plums and other fruit trees and pine species and are responsible for substantial economic losses in recent years. According to the statistics, in one small city alone, the area of the outbreak reached 8.34 km^2^. To control this outbreak, the local prevention and control agency organized 45 emergency technical training sessions, with more than 2200 participants. A total of 2.86 million CNY has been invested to mitigate the deleterious effects of this pest in its original range. We collected specimens from field surveys and identified them by female wings and male genitalia [12]. This species can also damage various plants and trees. Our field studies revealed that sex pheromone traps for spongy moths are equally attractive to this species. The implementation of various control measures has greatly reduced the densities of this species, but the potential detrimental consequences for quarantine of this species being mistaken for ASMs cannot be underestimated.

Several molecular markers and diagnostic tools have been developed to mitigate the spread of spongy moths. Correct identification is of greatest importance. As *Lymantria* species cannot be identified during the egg stage, accurate identification requires rearing individuals to adulthood and dissecting the genitalia. This process takes two to three months, which is not rapid enough for typical shipping schedules. This has necessitated the development of molecular methods that can be used to rapidly identify morphologically similar moth species. Although methods for the rapid detection of ASMs and related species have been explored in previous studies [30,38], the ASM-specific primers that have been previously used need to be redesigned for several reasons. First, previously designed ASM-specific primers are unable to identify ASMs in light of the high levels of differentiation and genetic variation recently detected among ASMs in China, as mentioned above. Second, previous studies have limitations in sampling, as some ASM samples were continuously cultivated in the laboratory for many generations, and these samples were inevitably adapted to the laboratory conditions, which may be different from those in the field under natural conditions. Although *Lymantria xylina* and *Lymantria monacha* are also on the alert list for North America, we have previously developed corresponding rapid detection techniques that have occurred in China in lesser numbers in recent years [30], so for the time being, we will only consider updating the ASM detection method.

One of the most effective techniques for species identification is species-specific COI (SS-COI) using species-specific primers. This technique permits species to be identified according to the presence or absence of specific bands on an agarose gel, and no sequencing is required [39]. Specifically, species are identified via gel electrophoresis of PCR-amplified products of known size [40]. Primer pairs are designed to permit the identification of specific lengths of the target species in a region of DNA shared among several similar species. These primers should only amplify sequences specific to the target species, and universal primers should amplify the DNA sequences shared among similar species. This approach has been used to rapidly identify several insect species [41,42,43,44].

Here, we improve the method for the rapid and accurate identification of ASMs using a pair of SS-COI primers. A pair of specific primers and a molecular detection system are established based on the COI gene. This method is simple to operate and takes only 2–3 h to complete. This improved method will aid the identification of ASMs in China and, thus, help prevent the spread of ASM to other regions.

## 2. Materials and Methods

### 2.1. Samples

ASM and other species of *Lymantria* similar to ASM in morphology were collected: *L. xylina*, *L. monacha*, *L. apicebrunnea*, *L. fumida*, *L. similis*, *Lymantria mathura* Moore, 1865, *Lymantria nebulosa* Wileman, 1910, and *Lymantria marginata* Walker, 1855. Pheromone traps were used to capture adults and egg masses were collected on the trunks of host trees. Eggs collected were reared in the laboratory on an artificial diet. All species were identified by reference to taxonomic keys [12,45]. Specific information collected is provided in Table 1. All adults and larvae collected were immersed in absolute ethanol and stored at –20 °C. The morphology of the spongy moth at several developmental stages is shown in Figure 1 and the morphology of the closely related species is shown in Figure 2.

### 2.2. Preparation of Template DNA

DNA was extracted from samples using the OMEGA D0926 Insect DNA Kit following the manufacturer’s instructions. First, approximately 20–50 mg of tissue was obtained; for adult moths, thorax or leg tissue was obtained. Leg tissue is often recommended for extracting genomic DNA from various types of insects to maintain the integrity of specimens. Following 3 days of starvation, anhydrous ethanol was used to kill larvae, and their abdominal tissues were extracted. Finally, DNA was eluted using 80 µL of elution buffer, the concentration and integrity of DNA samples were determined using a Nanodrop 8000 spectrophotometer and agarose gel electrophoresis was conducted with the PCR products.

### 2.3. DNA Barcoding and Phylogenetic Relationships of Lymantria

The barcode region of the *COI* gene was amplified using a pair of primers LCO1490 (5′-GGTCAACAAATCATAAAGATATTGG-3′) and HCO2198 (5′-TAAACTTCA GGGTGACCAAAAAATCA-3′) for species identification [46]. PCR was conducted in a reaction volume of 25 µL, with 12.5 µL of 2× Taq PCR mix (Zhongke Yubo), 1 µL each of the forward and reverse primers, 1 µL of DNA template and 9.5 µL of sterile distilled water. The thermal cycling conditions were as follows: pre-denaturation at 94 °C for 3 min, 30 cycles of denaturation at 94 °C for 30 s, annealing at 55 °C for 30 s, elongation at 72 °C for 1 min and a final extension at 72 °C for 5 min. After the PCR products were sequenced, they were compared against the Genbank database to identify species.

The phylogenetic relationships of *Lymantria* were subsequently determined based on DNA barcoding. Sequences for the other *Lymantria* species were downloaded from the National Center for Biotechnology Information (NCBI) database, accessed on 18 July 2022. (https://www.ncbi.nlm.nih.gov/) (Table 2). The neighbor-joining method was used to infer the phylogeny, 1000 bootstrap replicates were used to evaluate branch support and bootstrap values were shown next to branches in the phylogenetic tree [47]. The p-distance method was used to calculate genetic distances (number of base differences per site) [48].

### 2.4. Amplification of COI and the Design of Primers Specific for ASMs

A pair of universal primers, C1-J1709 and C1-N2776, designed in a previous study, were used to amplify the *COI* genes of ASMs [49]. For reasons of morphological similarity and proximity of kinship, we amplified the *COI* genes of the following four species: *L. dispar*, *L. xylina*, *L. monacha* and *L. apicebrunnea*. The sequences of these universal primers were as follows: C1-J1709: 5′-AATTGGWGGWTTYGGAAAYTG-3′ and C1-N2776: 5′-GGTAATCAGAGTATCGWCGNGG-3′. The thermal cycling conditions for the PCR reactions were the same as those described in 2.3.

Agarose gel electrophoresis (1.5%; 110 V, 25–30 min) was conducted using 5 µL of PCR products and a DNA marker (DL2000, Takara) was also added to the gel. Ultraviolet light was used to observe the electrophoresis products and samples with bright bands were sent to Qing Ke Co. (Beijing, China) for bidirectional sequencing. The sequences were then blasted against the NCBI database to determine whether the amplified products corresponded to the target bands. The sequences were analyzed using BioEdit 7.2.5 software. Sequences with primer sequences at each end, unstable sequences and inaccurate sequences were removed [50].

Comparisons of sequences were conducted within and between species, and regions that were conserved within ASMs but highly variable among species were used to design specific primers. Primer 5.0 was used to design specific primers for ASMs (Figure 3). ASMF: 5′-CCTTCTACTTTTATCTTTACCTGTT-3′ and ASMR: 5′-ATTGTAGCAGAGGTAAAG-3′.

### 2.5. Assessment of the Specificity and Sensitivity of the SS-COI PCR Assay

DNA extracted from four *Lymantria* species (Asian spongy moth, *L. xylina, L. apicebrunnea* and *L.monacha* ) was used as the template to determine whether ASMF/ASMR could be used to specifically identify ASMs, and this was verified according to whether gel electrophoresis revealed the presence of bands of expected sizes. A Control mix, the PCR mix without DNA, was used as a blank control. The primers used in the PCR system were ASMF/ASMR, and the thermal cycling conditions were as follows: pre-denaturation at 94 °C for 3 min, denaturation at 94 °C for 30 s, annealing at 53 °C for 30 s, elongation at 72 °C for 1 min and a final extension at 72 °C for 5 min. Agarose gel electrophoresis (1.5%) was performed using the PCR products, and a DNA marker (DL2000) was added to the gel.

We also conducted experiments aimed at determining the most optimal annealing temperatures for ASMF and ASMR. Eight annealing temperatures around the optimal annealing temperature (53 °C) recommended by Primer 5.0 software were tested, including 56 °C, 55 °C, 54 °C, 53 °C, 52 °C, 51 °C, 50 °C and 49 °C. PCR reactions were conducted as described above.

To test whether the primer pairs are suitable for all geographical populations in China, we tested the ability of these primers to detect ASMs from 14 regions. We also extracted DNA from single eggs, first, second, third, fourth, fifth, sixth instar, pupa and adults of ASM and added them to the PCR system with control mix as the blank control. The objective of these tests was to evaluate the utility of ASMF/ASMR for the identification of ASMs from different geographical populations and at various developmental stages.

Moths cannot be identified based on morphological characteristics when only part of a moth tissue is found at quarantine sites. We, thus, used two approaches to evaluate the sensitivity of ASMF/ASMR. First, we performed PCR using serial dilutions of ASM template DNA at concentrations of 30 ng, 20 ng, 30 pg, 20 pg and 300 fg. Mixed samples containing DNA from *L. xylina* (a species closely related to ASMs) and ASMs were also used as templates to test the sensitivity of the primers, and the ratios of the DNA templates of ASMs and *L. xylina* used were as follows: (1:0), (1:1), (1:10), (1:50), (1:100) and (1:1000).

### 2.6. Random Detection of Moths in the Wild with ASMF/ASMR

Other *Lymantria* species in the wild were collected through light traps to evaluate the efficacy of the ASMF/ASMR for identification of ASMs in random insect samples. After insects were collected, PCR reactions and electrophoresis were conducted using the genomic DNA of *Lymantria fumida*, *Lymantria similis*, *Lymantria mathura*, *Lymantria nebulosa* and *Lymantria marginata* as templates. The procedures for DNA extraction and PCR were the same as those described above.

## 3. Results

### 3.1. DNA Barcoding and Phylogenetic Analysis of Lymantria

Our identifications made according to BLAST searches against the Genbank database were consistent with those based on morphology. The *COI* barcode fragments amplified from the collected samples and the sequences of other *Lymantria* species downloaded from the NCBI database were used to build phylogenetic trees (Figure 4) and genetic distance matrices (Figure 5). It can be seen that *L. apicebrunnea* and *Lymantria schaeferi* (Schintlmeister, 2004) are clustered together and their genetic distance based on the COI barcode fragment is 0. This finding is consistent with the results of the BLAST comparison, which indicated that these two species are closely related. The genetic distance between the two ASM subspecies, *Lymantria dispar asiatica* and *Lymantria dispar japonica*, based on our selected fragment, was 0. The genetic distance between the two species *Lymantria albescens* (Hori and Umeno, 1930) and *Lymantria postalba* (Inoue, 1956) in Japan was also 0. The species most closely related to *L. dispar,* according to our data, was *Lymantria umbrosa* (genetic distance of 0.02 to *Lymantria dispar*). The genetic distances of several species in our analysis, including *L. schaeferi*, *L. xylina* and *L. apicebrunnea*, to *L. dispar* ranged from 0.06 to 0.07.

### 3.2. Amplification of the COI Gene Using Generic Primers

DNA from *L. dispar*, *L. xylina*, *L. monachal* and *L. apicebrunnea* was used as templates for amplifying the same fragment of the mitochondrial *COI* gene using the universal primers C1-J1709 and C1-N2776. Sequence alignment following bidirectional sequencing of the PCR products also indicated that this pair of primers amplified the same target fragment (see Appendix A).

### 3.3. Primer Specificity Test

PCR and agarose gel electrophoresis were conducted using the DNA of the four species (*L. dispar*, *L. xylina*, *L. monacha* and *L. apicebrunnea*) as templates. Bright bands on the gel were only observed in the lanes corresponding to ASM samples (Figure 6). Next, we bidirectionally sequenced the PCR products that generated bands, and a blast search revealed that the products amplified by the specific primers belonged to the target species, suggesting that these primers were specific for ASMs.

We tested eight annealing temperatures (56 °C, 55 °C, 54 °C, 53 °C, 52 °C, 51 °C, 50 °C and 49 °C) to determine the optimal annealing temperature range of the pair of designed primers. Bright bands were observed at all eight annealing temperatures (Figure 7). Next, we tested the following annealing temperatures below 49 °C and above 56 °C: 47 °C, 48 °C, 57 °C, 58 °C, 59 °C and 60 °C. None of these annealing temperatures were suitable for the amplification of ASMF and ASMR, as no bands were observed (see Appendix A). Thus, the most suitable annealing temperatures for ASMF/ASMR were 49–56 °C.

ASM samples from 14 regions in China where ASMs are known to occur were used to evaluate whether these primers could be used for the identification of ASMs from various regions (Figure 8). We also evaluated the ability of the primers to identify ASMs at various developmental stages from single eggs to adults. Bright bands were observed in all gels (Figure 9), suggesting that the primers could be used for the identification of ASMs at various developmental stages.

### 3.4. Sensitivity Test of the ASM-Specific Primers

To determine the sensitivity of the ASMF/ASMR primers, we carried out serial dilutions of DNA from ASMs. Bands were produced when reactions contained 30 ng, 20 ng and 30 pg of template DNA, suggesting that the minimum amount of DNA required to detect ASMs using these primers is 30 pg of DNA template (Figure 10a). We also evaluated mixed samples containing DNA templates of spongy moths and *L. xylina* to evaluate the sensitivity of the ASMF/ASMR primers. ASMs were detected at all ratios of template DNA, with the exception of 1:1000, suggesting that the primers were sufficiently sensitive for identifying ASMs (Figure 10b).

### 3.5. Random Detection of Moths in the Wild with ASMF/ASMR

The specificity of the designed primers was also evaluated using randomly captured moths in the field, and experiments were conducted using five other common *Lymantria* species. Bands were only observed in lanes with spongy moth DNA (Figure 11).

## 4. Discussion

Continuous growth in global economic trade has promoted the spread of quarantine pests and biological invasions can disrupt the ecology of local ecosystems. There is, thus, a need for stringent quarantine measures to be implemented. The accurate identification of quarantine pests is the first step required to determine whether a quarantine should be implemented, as this is necessary to ensure that the appropriate actions are being taken. Accurate and rapid techniques that permit the identification of pests on a country’s quarantine pest list are essential for achieving this goal. The development of such methods is a special challenge for insects, as the taxonomic relationships among many insects are complex. Moreover, many closely related insect species show high similarity in morphology. Misidentifications can also impede trade and economic development when non-quarantine pests are mistaken for quarantine pests. In some cases, misidentifications can result in the excessive use of chemical pesticides when they are not needed and, thus, cause indirect damage to the ecological environment, given that not every insect species requires human intervention to be controlled. Non-quarantine pests, in particular, have many natural enemies that keep their populations in check. Repeated misidentifications can amplify the deleterious effects of implementing improper control measures.

### 4.1. Development of Rapid and Accurate Molecular Detection Techniques Is a Necessity

Rapid and accurate methods for identifying quarantine pest species are necessary for studies examining their behavior, ecology and physiology. These methods are also important for preventing the spread of these pests to new areas. Most quarantine insect pests are currently identified based on morphological characteristics. However, there are limitations associated with traditional morphology-based identification approaches. For example, identification of insects is difficult when only part of the tissue is available or when insects are in their early life stages. These traditional methods are error-prone and time-consuming, and this can affect the ability to accurately detect quarantine pests and prevent future invasions. Lepidopteran insects are not only diverse and similar in morphology but their scales can easily fall off. Unless specific measures are taken, the scales are often not intact under normal circumstances, and this can affect the reliability of morphology-based identification approaches. Given that insect species vary substantially in male genitalia, male genitalia have been frequently used for insect identification. Many methods for collecting and measuring external genitalia have been developed for various insects, especially Lepidopterans [51,52]. However, dissection of the external genitalia and morphological comparison by microscopy is time consuming and requires specialized technicians, while time-sensitive methods are needed at quarantine ports. Mitochondrial DNA markers that are polymorphic within and among species and populations are also effective for species identification [53,54].

The *COI* gene has been shown to be effective for the identification of various animal species. In 2003, Hebert first proposed that the *COI* gene could be used for species identification. Since then, DNA barcoding technology has received wide research attention [55,56,57]. DNA barcode technology is now widely used for species identification and in studies of molecular evolution, genetic variation and biodiversity. Some of the reasons for its wide use include its low cost, ease of implementation and robustness to interference from the external environment. DNA barcoding with *COI* genes has been shown to be effective for the identification of various animal species [58]. In the animal kingdom, more than 95% of the species can be accurately identified at the species level using this approach [59]. Kang used DNA barcodes to detect *Lymantria* species in Korea and showed that this approach can distinguish between closely related species [60].

### 4.2. For Practical Reasons, SS-COI PCR Is a Highly Efficient and Suitable Method for Ports

There are several advantages of the SS-COI PCR technique based on the mitochondrial *COI* gene, including its high reproducibility, and this makes it an effective approach for the large-scale and rapid identification of insects. As discussed in the Introduction, our study was motivated by the discovery of high levels of genetic variation among ASMs in China. ASMs are native to China, and patterns of genetic variation in this species likely change continuously with global trade activities. To mitigate the spread of ASMs to other countries, Chinese quarantine agencies have implemented stringent quarantine measures. However, the recent discovery of high levels of genetic variation in ASMs has necessitated the development of a new approach for the rapid detection of ASMs. Many advanced methods have been developed for moth identification, such as DNA barcoding, loop-mediated isothermal amplification (LAMP) technology and recombinase polymerase amplification (RPA) technology, and the most widely used method for the rapid identification of animals is DNA barcoding. Although DNA barcoding has often been used for the identification of economically important insects, one limitation of this method is that it requires a DNA barcode database with a sufficient number of species to make accurate identifications [61]. Second, LAMP technology and RPA technology have been widely used because of their high sensitivity. However, the high sensitivity and amplification rates of LAMP can increase the rate of false positives. The loop primers used in LAMP increase the amplification efficiency but are vulnerable to generating false positives because of the non-specific pairing of the loop primers [62]. The RPA technique is an effective species-identification approach that does not require a thermal cycler, but the partial non-specific amplification and false positives associated with reactions conducted at a constant temperature are major shortcomings that merit consideration. In addition, the TwistAmp^®^ Basic Kit required for RPA is expensive, is not suitable for large-scale testing in ports and it is not available in some remote locations [41]. In light of these considerations, we used SS-COI PCR because our aim was to explore the utility of an approach that could be used in real-world quarantine situations. We showed that our new primers were highly sensitive and specific and, thus, that they could be used for large-scale testing in ports. We believe that the SS-COI PCR is a cost-effective approach with high specificity and sensitivity that would be a convenient tool for quarantine agencies that do not have easy access to well-equipped laboratories.

### 4.3. The Updated ASM Detection Method has Enough Specificity and Sensitivity for Application

We collected samples of ASMs throughout the range of this species in China and designed ASM-specific primers. We also re-evaluated the phylogenetic relationships of *Lymantria* using these samples. Our findings revealed that *L. postalba* and *L. albescens* are closely related, with a genetic distance between them based on COI fragments of 0. The taxonomic status of the two species remains controversial, with some researchers arguing that “*L. albescens* and *L. postalba* are at best two forms of a single species” [63]. Similarly, we found that the genetic distance between *L. apicebrunnea* and *L. schaeferi* based on COI fragments was also 0. This discovery has raised questions regarding the taxonomic status of these two species. As we did not have specimens of *L. schaeferi* and our phylogenies were constructed using only *COI* fragments, additional research is needed to re-evaluate the taxonomic status of these two species. What is clear from our data is that *L. apicebrunnea* and *L. dispar* are closely related. Their high morphological similarity and ability to feed on the same hosts mean that special efforts should be made to prevent the spread of *L. apicebrunnea* so that it does not become a major pest. On the other hand, our tests showed that our ASM-specific primers have high sensitivity and specificity and, thus, would be effective for the rapid detection of ASMs in quarantine ports. Compared to previous studies, the ASM-specific primers that we designed are effective for detecting ASM samples from all geographic populations in China and developmental stages, including from the first instar to the last instar stages, which was not performed in previous studies. Furthermore, the specific primers we designed are effective under a wider range of annealing temperatures (49–56 °C) compared with the primers used in previous studies. The results of previous studies may have involved only one suitable annealing temperature. In such a scenario, the replacement of the PCR instrument during detection might lead to detection errors due to temperature differences among PCR instruments. The ability of our primers to function effectively under a wide range of annealing temperatures can eliminate the effects of temperature differences among PCR instruments and enhance the accuracy of detecting the target species. We added different proportions of ASM DNA to samples of DNA of other moths to evaluate the sensitivity of our primers and the results indicated that the primers could accurately detect ASMs at a ratio as low as 1:100. SS-COI PCR using these primers is, thus, a rapid and accurate method for identifying ASMs. Given that our primers were designed for ASMs, additional work is needed to develop EGM-specific primers. Over the last decade, populations of other *Lymantria* species have been maintained at low levels, and these species have not yet induced major economic losses. Due to the high diversity in *Lymantria* species and limitations in available samples, the specificity of our primers for ASMs relative to other *Lymantria* species requires testing. Spongy moths are the most destructive pests in the genus *Lymantria*, and this was our motivation for redesigning spongy-moth-specific primers. In the future, multiple PCR primers should be designed to identify many species. In addition, if conditions permit, more accurate and sensitive technologies, such as RPA and LAMP, should be developed for the detection of invasive species and reduce trade losses.

## 5. Conclusions

Invasions of ASMs are considered more serious threats to forest resources than invasions of ESMs because of their ability to induce greater amounts of damage. The implementation of quarantine measures is essential to prevent the spread of ASMs. ASMs are native to and widely distributed in China. We extensively sampled ASMs in China and used SS-COI PCR to rapidly and accurately identify ASMs. This method is an improvement on existing methods and could be used in quarantine laboratories. It could, thus, be useful for monitoring and preventing the spread of ASMs to other regions.

## Figures and Tables

**Figure 1 insects-14-00094-f001:**
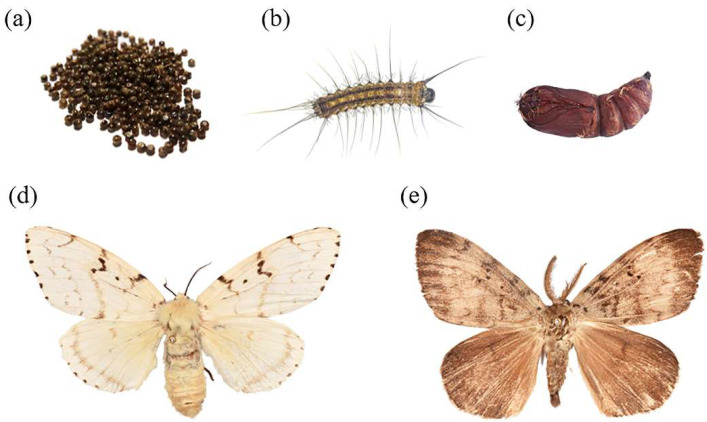
ASMs at four developmental stages. (**a**): Egg. (**b**): Larva. (**c**): Pupa. (**d**): Female adult. (**e**): Male adult.

**Figure 2 insects-14-00094-f002:**
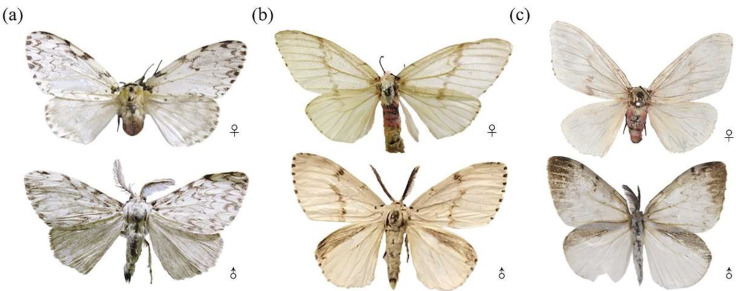
The species closely related to spongy moth used as controls. Moths in the top row are female adults and moths in the bottom row are male adults. (**a**): *Lymantria monacha*. (**b**): *Lymantria xylina*. (**c**): *Lymantria apicebrunnea*.

**Figure 3 insects-14-00094-f003:**
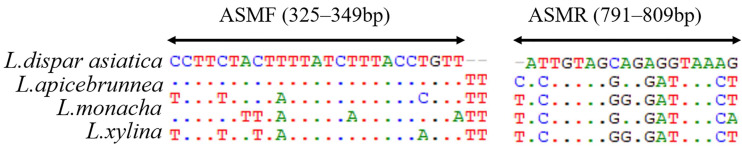
Alignment of COI gene sequences of four species. The ASMF (**left**) and ASMR (**right**) primer sequences are located at the regions in the COI genes showing high variation among *Lymantria* species but low variation within ASMs.

**Figure 4 insects-14-00094-f004:**
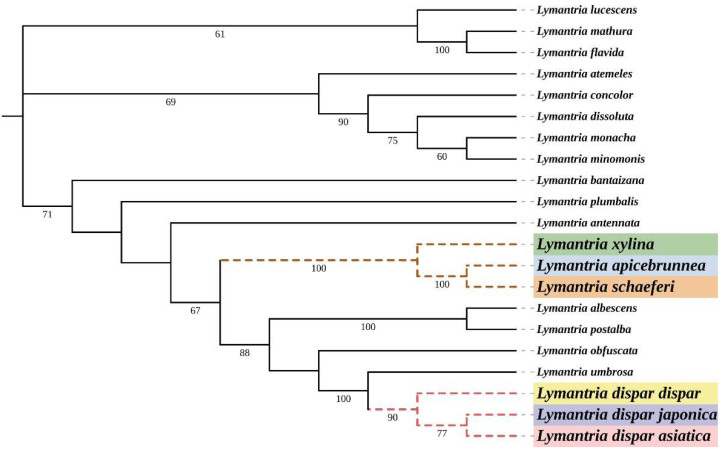
DNA barcode-based neighbor-joining phylogenetic tree of *Lymantria*.

**Figure 5 insects-14-00094-f005:**
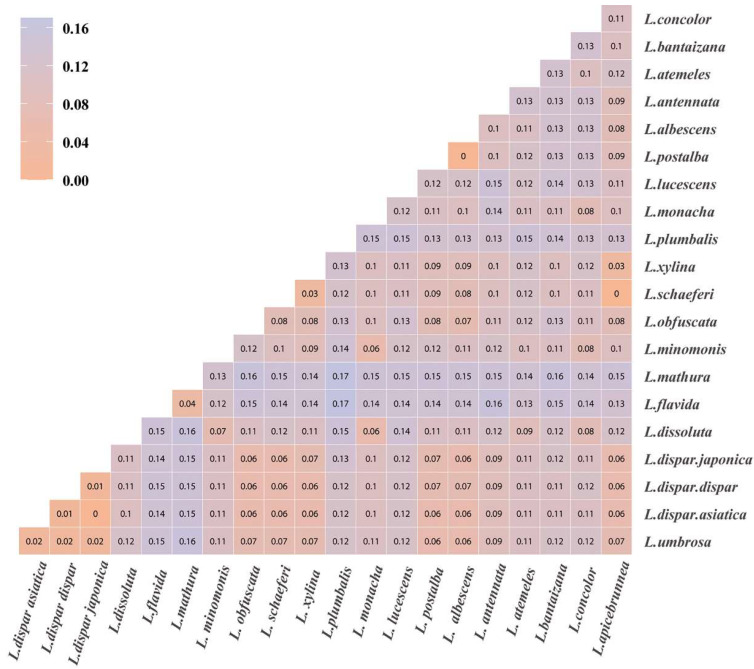
Genetic distance heatmap of *Lymantria*. The shade of color indicates the genetic distance; orange indicates low genetic distances and blue indicates high genetic distances.

**Figure 6 insects-14-00094-f006:**
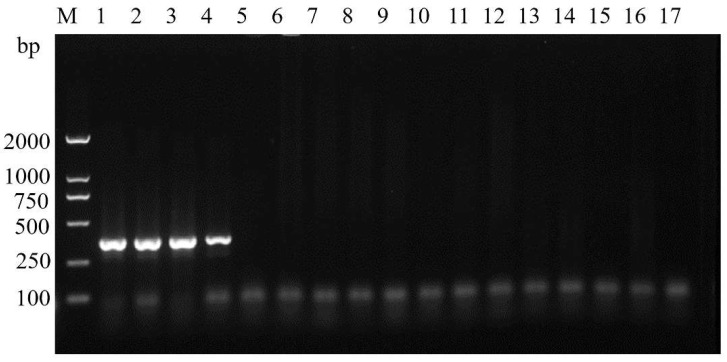
Agarose gel electrophoresis showing specificity of the ASM-specific primers ASMF/ASMR. M: DL2000 DNA marker (from top to bottom: 2000 bp, 1000 bp, 750 bp, 500 bp, 250 bp and 100 bp); Lanes 1–4: Asian spongy moth (1–3 are *Lymantria dispar asiatica*: Ulanhot, Cheng du and Guizhou; 4: *Lymantria dispar japonica*: Honshu); Lanes 5–8: *Lymantria xylina*; Lanes 9–12: *Lymantria monacha*; Lanes 13–16: *Lymantria apicebrunnea*; and Lane 17: Control mix (the PCR mix without DNA).

**Figure 7 insects-14-00094-f007:**
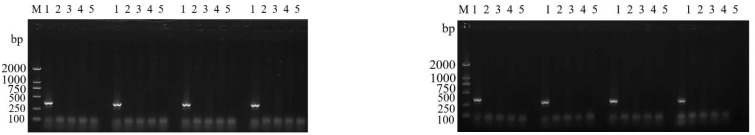
Agarose gel electrophoresis of the most suitable annealing temperature for ASMF/ASMR. The eight groups correspond to different annealing temperatures. Left to right: 56 °C, 55 °C, 54 °C, 53 °C, 52 °C, 51 °C, 50 °C and 49 °C. M: DL2000 DNA marker; Lane 1: *Lymantria dispar asiatica*; Lane 2: *Lymantria xylina*; Lane 3: *Lymantria monacha*; Lane 4: *Lymantria apicebrunnea*; Lane 5: Control mix (the PCR mix without DNA).

**Figure 8 insects-14-00094-f008:**
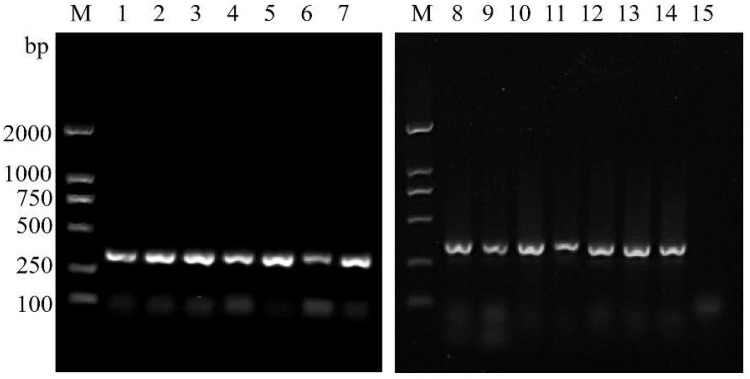
Agarose gel electrophoresis of ASMF/ASMR for specificity tests of different geographical populations. M: DL2000 DNA marker; Lanes 1–15: ASMs from different geographic regions. From left to right: Tianjin, Hegang, Shandong, Chifeng, Chengde, Hebei, Charisu, Tongliao, Beijing, Liaoning, Shaanxi, Shuozhou, Yuncheng, Inner Mongolia, and control mix (the PCR mix without DNA).

**Figure 9 insects-14-00094-f009:**
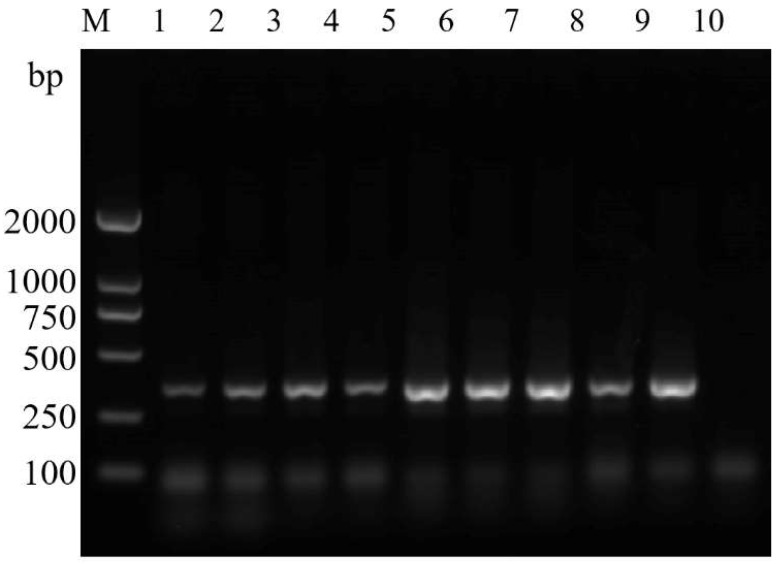
Agarose gel electrophoresis of ASMF/ASMR for specificity tests of different developmental stages of Asian spongy moth. M: DL2000 DNA marker; Lanes 1–10: ASM egg, first instar, second instar, third instar, fourth instar, fifth instar, sixth instar, pupa, adult and control mix (the PCR mix without DNA).

**Figure 10 insects-14-00094-f010:**
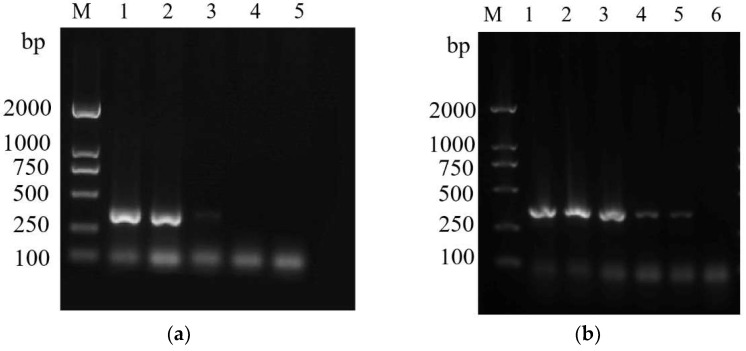
Test of the sensitivity of ASMF/ASMR for the identification of ASMs. (**a**) Agarose gel electrophoresis of a series of dilutions of template DNA. M: DL2000 DNA marker; Lanes 1–5: 30 ng/μL, 20 ng/μL, 30 pg/μL, 20 pg/μL, and 300 fg/μL DNA from Asian spongy moth, respectively; (**b**) Agarose gel electrophoresis for mixed samples containing ASM and *Lymantria xlina* template DNA. M: DL2000 DNA marker; Lanes 1–6: Ratio of the DNA templates of the two species: (1:0), (1:1) (1:10), (1:50), (1:100) and (1:1000), respectively.

**Figure 11 insects-14-00094-f011:**
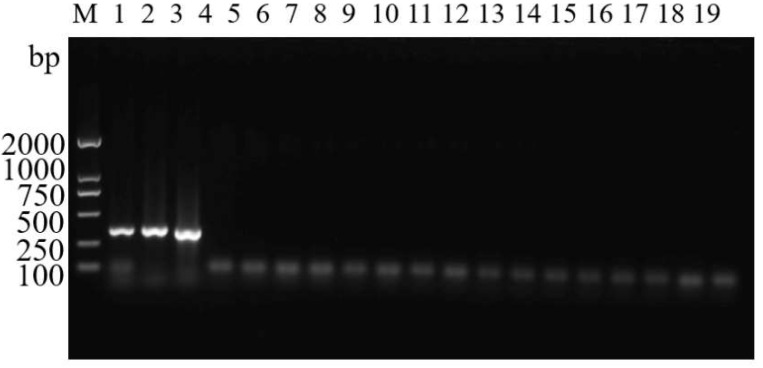
Agarose gel electrophoresis of randomly collected ASM samples and samples of other *Lymantria* species from insect-trapping lamps. M: DL2000 DNA marker; Lanes 1–3: ASM; Lanes 4–6: *Lymantria fumida*; Lanes 7–9: *Lymantria similis*; Lanes 10–12: *Lymantria mathura*; Lanes 13–15: *Lymantria nebulosa*; Lanes 16–17: *Lymantria marginata*; and Lanes 18–19: control mix (the PCR mix without DNA).

**Table 1 insects-14-00094-t001:** Collection information of species used in this study to design and validate species-specific primers for important quarantine pest Asian spongy moth to prevent misidentification.

Species No.	Species	Location	Developmental Stage	Date
1	*Lymantria dispar asiatica*	Tian Jing	Adult and eggs	August, 2021
2	*Lymantria dispar asiatica*	He Gang	Adult and eggs	August, 2021
3	*Lymantria dispar asiatica*	Shan Dong	Adult and eggs	July, 2021
4	*Lymantria dispar asiatica*	Chi Feng	Adult and eggs	July, 2021
5	*Lymantria dispar asiatica*	Cheng De	Adult and eggs	July, 2021
6	*Lymantria dispar asiatica*	He Bei	Adult and eggs	June, 2020
7	*Lymantria dispar asiatica*	Cha Ri Su	Adult and eggs	September, 2020
8	*Lymantria dispar asiatica*	Tong Liao	Adult and eggs	August, 2021
9	*Lymantria dispar asiatica*	Bei Jing	Adult and eggs	August, 2020
10	*Lymantria dispar asiatica*	Liao Ning	Adult and eggs	September, 2021
11	*Lymantria dispar asiatica*	Shaan Xi	Adult and eggs	July, 2021
12	*Lymantria dispar asiatica*	Shuo Zhou	Adult and eggs	August, 2021
13	*Lymantria dispar asiatica*	Yun Cheng	Adult and eggs	July, 2021
14	*Lymantria dispar asiatica*	Inner Mongolia	Adult	September , 2021
15	*Lymantria dispar asiatica*	Gui zhou	Adult	June, 2020
16	*Lymantria dispar asiatica*	Chengdu	Adult	July, 2020
17	*Lymantria dispar asiatica*	Ulanhot	Adult	September, 2020
18	*Lymantria dispar japonica*	Japan	Adult	September, 2017
19	*Lymantria monacha*	Inner Mongolia	Adult and eggs	August, 2019
20	*Lymantria xylina*	Fu Jian	Adult	August, 2019
21	*Lymantria apicebrunnea*	Yun Nan	Adult and eggs	July, 2021
22	*Lymantria fumida*	Gui Zhou	Adult	July, 2022
23	*Lymantria mathura*	Si chuan	Adult	August, 2022
24	*Lymantria nebulosa*	Yun Nan	Adult	August, 2022
25	*Lymantria marginata*	Yun Nan	Adult	August, 2022
26	*Lymantria similis*	Yun Nan	Adult	August, 2022

**Table 2 insects-14-00094-t002:** COI sequences of *Lymantria* species downloaded from the NCBI database for reconstructive phylogenetic relationships.

No.	Species	Accession No.
1	*Lymantria umbrosa*	HM775854
2	*Lymantria dissoluta*	HM775756
3	*Lymantria flavida*	HM775761
4	*Lymantria mathura*	HM775782
5	*Lymantria minomonis*	HM775790
6	*Lymantria obfuscata*	HM775826
7	*Lymantria schaeferi*	HM775840
8	*Lymantria xylina*	MW085568
9	*Lymantria plumbalis*	HM775836
10	*Lymantria monacha*	HM875343
11	*Lymantria lucescens*	LC406220
12	*Lymantria postalba*	LC406195
13	*Lymantria albescens*	LC406184
14	*Lymantria antennata*	HQ921458
15	*Lymantria atemeles*	KP759547
16	*Lymantria bantaizana*	KX436537
17	*Lymantria concolor*	KX436392
18	*Lymantria dispar dispar*	HM418030
19	*Lymantria dispar japonica*	KY923061

## Data Availability

Publicly available data sets were analyzed in this study. These data can be found in NCBI database (https://www.ncbi.nlm.nih.gov/, accessed on 18 July 2022). The accession numbers of *Lymantria dispar asiatica* are OP700782-OP700784; the accession numbers of *Lymantria apicebrunnea* are OP700785-OP700787; the accession numbers of *Lymantria xylina* are OP700790-OP700791; the accession numbers of *Lymantria monacha* are OP700788-OP700789. This process can be initiated upon request to the corresponding author.

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
