# Peer review of "Improvement in the Identification Technology for Asian Spongy Moth, Lymantria dispar Linnaeus, 1758 (Lepidoptera: Erebidae) Based on SS-COI"

_insects, 2023, doi:10.3390/insects14010094_

Round 1

Reviewer 1 Report

Introduction section:

comment 1: A justification is needed on how the authors innovate in this technique compared to previous works. For example, doi: 10.3390/insects10050146.

comment 2: In lines 153 to 156; there are other molecular techniques such as DNA barcoding to identify species, and not necessarily by PCR-Sanger sequencing. the authors should expand and justify this topic.

Materials and Methods section:

comment 3: in the sample section and Table 1; the authors indicate that adults and eggs were collected. As the authors confirmed that the eggs are from a specific moth species and not from another, since there is a high similarity in egg morphology between the species?

comment 4: In the line 183, these Moth species were identified by which taxonomic keys?

comment 5: In the line 183, the authors mention that it was by performing DNA barcoding analysis; but, how was this methodology developed? please describe in materials and methods and results.

comment 5: In section 23 to 2.5; It is not clear how subsequent PCRs were performed for the final analysis. Different PCR amplifications (What is the size in bp of the amplified and sequenced fragment? How was it sequenced?) are described, first with universal primers (2.3. Phylogenetic relationships of Lymantria); then a second set of universal primers (2.4. Amplification of COI and the design of primers specific for ASMs) and then a third set of primers (specific primers for ASMs). Did each PCR result from the previous one? If so, what was the methodology used? The writing of these sections is not clear on this issue

comment 6: In the line 269, what kind of Complex patterns of genetic differentiation? are these deletions and duplications?

Results section:

comment 7: the results do not show a significant contribution, since they describe the development of successive PCRs. on the other hand, it is not clear how the phylogenetic and genetic distance analyses of the genus contribute significantly to the objective set by the authors.

Discussion section:

comment 8: This section lacks a discussion of the efficiency of the technique and how it differs from previous molecular techniques. A detailed comparison of the results obtained (technically) with those of many previous authors seeking to validate the results is lacking.

comment 9: What is the conclusion of the work?

Author Response

Dear Reviewer,

On behalf of my co-authors, we thank you very much for giving us an opportunity to revise our manuscript, and we appreciate very much for your positive and constructive comments and suggestions on our manuscript entitled “Improvement of the identification technology for Asian Spongy Moth, Lymantria dispar Linnaeus, 1758 (Lepidoptera: Erebidae) based on SS-COI”. We have studied the comments carefully and have made revisions marked in highlight (The blue highlight is a language improvement that native English speakers help us make. It does not change the original meaning of the sentence, but merely adjusts the grammar and the order of the words to more closely resemble the English expression. The yellow highlighting is our point-to-point revision based on the reviewers' comments). Attached please find the revised version, which we would like to submit for your kind consideration.

We tried our best to improve the manuscript and made some changes in the manuscript. These changes will not affect the content and framework of the paper. We hope that our revision could address all of the questions by the reviewers.

Thank you very much for your considering our manuscript for publication. We are looking forward to hearing from you soon.

Sincerely yours,

Prof. Juan Shi

Beijing Key Laboratory for Forest Pest Control, Beijing Forestry University,Sino-French Eurasian Forest Invasive Biology Laboratory, Beijing Forestry University

P.O. Box 113, 35 Qinghua East Road, Haidian District, Beijing 100083, P. R. China

Tel: +86-10-62336423

Fax: +86-10-62336423

Response to Reviewer 1 Comments

Point 1: A justification is needed on how the authors innovate in this technique compared to previous works. For example, doi: 10.3390/insects10050146.

Response 1: Thank you for your careful review of the manuscript, but it is possible that our unstandard language may have caused you confusion. Regarding the paper you mentioned: doi: 10.3390/insects10050146, it is a study that explores the genetic structure and population history of Lymantria dispar in China and surrounding areas based on COI genes. In contrast, our paper is an attempt to improve the rapid identification of Asian spongy moth. Although both use COI genes, genetic structure and rapid identification are two different research methods, and we are not improving on genetic structure, but rather building on previous studies of rapid detection. We did this study because the molecular detection technique is based on nucleic acid sequences and, as we mentioned in the introduction, our previous study found that Asian spongy moth in China showed unexpected genetic divergence, that is, divergence in nucleic acid sequences. Therefore, we believe that, based on the new genetic divergence identified, it is necessary to improve the existing rapid detection technique for Asian spongy moth to enable a more comprehensive and accurate identification of ASM. This is why we cite this reference (doi: 10.3390/insects10050146), which also demonstrates the genetic differentiation of ASM in China.

Considering the structure of the manuscript, we supplement the innovation of our technology and previous works, details as follows: “Compared to previous studies, the ASM-specific primers that we designed are effective for detecting ASM samples from all geographic populations in China and developmental stages, including from the first instar to the last instar stages, which was not performed in previous studies. Furthermore, the specific primers we designed are effective under a wider range of annealing temperatures (49–56°C) compared with the primers used in previous studies. The results of previous studies may have involved only one suitable annealing temperature. In such a scenario, the replacement of the PCR instrument during detection might lead to detection errors due to temperature differences among PCR instruments. The ability of our primers to function effectively under a wide range of annealing temperatures can eliminate the effects of temperature differences among PCR instruments and enhance the accuracy of detecting the target species. We added different proportions of ASM DNA to samples of DNA of other moths to evaluate the sensitivity of our primers, and the results indicated that the primers could accurately detect ASMs at a ratio as low as 1:100. ” (line 484-497)

Point 2: In lines 153 to 156; there are other molecular techniques such as DNA barcoding to identify species, and not necessarily by PCR-Sanger sequencing. the authors should expand and justify this topic.

Response 2: We apologize for our error in writing this paragraph, which was supposed to be a description of "SS-COI technology" and corresponds to the title "Improvement of the identification technology for Asian Spongy Moth, Lymantria dispar Linnaeus, 1758 (Lepidoptera: Erebidae) based on SS-COI". Although they stand for the same thing, they are not explained very clearly here due to our mistake. We would like to explain that, as you have suggested, there are other molecular techniques such as to identify species. we have only described the principles and methods of the SS-COI technique in detail because this is the technique used in our study and we thought it might be effective in explaining its principles in detail to help the reader understand our study. There are many other molecular identification techniques, such as DNA barcoding, RPA,LAMP, but given the length and subject matter of the manuscript, we have not expanded on all of them. We have therefore expanded on this theme in discussion 4.2: “Many advanced methods have been developed for moth identification, such as DNA barcoding, loop-mediated isothermal amplification (LAMP) technology, and recombinase polymerase amplification (RPA) technology, and the most widely used method for the rapid identification of animals is DNA barcoding. Although DNA barcoding has often been used for the identification of economically important insects, one limitation of this method is that it requires a DNA barcode database with a sufficient number of species to make accurate identifications [61]. Second, LAMP technology and RPA technology have been widely used because of their high sensitivity. However, the high sensitivity and amplification rates of LAMP can increase the rate of false positives. The loop primers used in LAMP increase the amplification efficiency but are vulnerable to generating false positives because of the non-specific pairing of the loop primers [62]. The RPA technique is an effective species identification approach that does not require a thermal cycler, but the partial non-specific amplification and false positives associated with reactions conducted at a constant temperature are major shortcomings that merit consideration. In addition, the TwistAmp® Basic Kit required for RPA is expensive, it is not suitable for large-scale testing in ports and it is not available in some remote locations [41]. In light of these considerations, we used SS-COI PCR because our aim was to explore the utility of an approach that could be used in real-world quarantine situations.” (line 445-463)

Point 3: in the sample section and Table 1; the authors indicate that adults and eggs were collected. As the authors confirmed that the eggs are from a specific moth species and not from another, since there is a high similarity in egg morphology between the species?

Response 3: We gratefully appreciate for your valuable suggestion. We are sorry that we did not give a detailed description of the sampling here. As shown in Table 1, we collected adults and eggs of Lymantria monacha, Lymantria apicebrunnea and Lymantria dispar asiatica. For other species, only adults were collected. This is due to the fact that Lymantria dispar asiatica and Lymantria apicebrunnea have been maintained at high levels in China in recent years. So we could easily collected both eggs and adults. However, due to the low occurrence density of other species in China recently, and the concealed location of eggs, we did not find eggs in the field collection process. Although Lymantria monacha has not occurred in large areas in recent years, we found eggs by accident. What we need to declare is that whether we collect eggs has no influence on our experiment. In Method 2.1, we have supplemented the sampling process in detail as follows: “ASM and other species of Lymantria are collected, which are similar to ASM in morphology: L. xylina, L. monacha, L. apicebrunnea, L. fumida, L. similis, L. mathura Moore, 1865, L. nebulosa Wileman, 1910, L. marginata Walker, 1855. Pheromone traps were used to capture adults and egg masses were collected on the trunks of host trees.”(line 174-178).

Point 4: In the line 183, these moth species were identified by which taxonomic keys?

Response 4: Thank you for pointing out this problem in manuscript. We identified the species by referring to the morphological description and taxonomic keys in Pogue&Schaefer (2007) and Zhao (2003). In addition we have added citations to relevant references in the manuscript (line 179).

Point 5:  In the line 183, the authors mention that it was by performing DNA barcoding analysis; but, how was this methodology developed? please describe in materials and methods and results.

Response 5: Thank you for pointing out this problem in the manuscript. According to your advice, We have added a more detailed description of the DNA barcoding in materials and methods and results. The details as follows: “ The barcode region of the COI gene was amplified using a pair of primers LCO1490 (5′-GGTCAACAAATCATAAAGATATTGG-3′) and HCO2198 (5′-TAAACTTCA GGGTGACCAAAAAATCA-3′) for species identification [46]. PCR was conducted in a reaction volume of 25 µL, with 12.5 µL of 2× Taq PCR mix (Zhongke Yubo), 1 µL each of the forward and reverse primers, 1 µL of DNA template, and 9.5 µL of sterile distilled water. The thermal cycling conditions were as follows: pre-denaturation at 94°C for 3 min, 30 cycles of denaturation at 94°C for 30 s, annealing at 55°C for 30 s, elongation at 72°C for 1 min, and a final extension at 72°C for 5 min. After the PCR products were sequenced, they were compared against the Genbank database to identify species.”(line 205-213)

“Our identifications made according to BLAST searches against the Genbank database were consistent with those based on morphology.”(line287-288)

Point 6: In section 2.3 to 2.5; It is not clear how subsequent PCRs were performed for the final analysis. Different PCR amplifications (What is the size in bp of the amplified and sequenced fragment? How was it sequenced?) are described, first with universal primers (2.3. Phylogenetic relationships of Lymantria); then a second set of universal primers (2.4. Amplification of COI and the design of primers specific for ASMs) and then a third set of primers (specific primers for ASMs). Did each PCR result from the previous one? If so, what was the methodology used? The writing of these sections is not clear on this issue

Response 6: We apologise for the confusion caused here. We would like to explain that although all PCRs are performed based on the COI gene, different primers amplify different fragments of the COI gene. Three primer pairs were used for PCR system and that their results are not correlated because they have different purposes. Moreover, all sequencing was bidirectional sequencing, which was done by a Beijing company (Ching Ke Co ).

In method 2.3,we aim at species identification based on DNA barcoding and the construction of Lymantria phylogenetic relationships. We use the primers LCO1490/HCO2198, which are documented in the literature to amplify Lepidoptera barcodes, and after PCR reactions and sequencing, we use them to identify species and construct phylogenetic relationships by comparison with the Genebank database. Here, this primer pair amplifies a fragment of 39bp to 696bp of the COI gene.

In method 2.4, we aimed to design primers specific to the ASM, and this PCR used another pair of primers C1-J1709 and C1-N2776 to amplify a 354bp-1145bp fragment of the COI gene. The amplified fragments from the four species were then subjected to bidirectional sequencing using ClustalW in BioEdit to find specific regions of ASM that differed from other species. A pair of 350 bp primers ASMF/ASMR (Figure 3) was designed to specifically identify ASM based on this region.

In method 2.5, we worked on testing whether ASMF/ASMR has a role in the specific identification of ASM. Here, the primers ASM/ASMR were used to perform the PCR, and gel electrophoresis is used to observe whether there are specific bands.

Point 7: In the line 269, what kind of Complex patterns of genetic differentiation? are these deletions and duplications?

Response 7: Thank you for pointing out this problem in manuscript. After we carefully checked the manuscript, we are repeated here which has been described in the introduction. We have deleted this sentence from the revised manuscript.

Point 8: the results do not show a significant contribution, since they describe the development of successive PCRs. on the other hand, it is not clear how the phylogenetic and genetic distance analyses of the genus contribute significantly to the objective set by the authors.

Response 8: We feel sorry for the inconvenience brought to you.We would like to explain that the aim of our experiment was to develop a method for the specific identification of ASM. Our approach was to design a pair of ASM-specific primers and to demonstrate that the primers were available by means of a specificity test and a sensitivity test. That is, the primers can specifically identify ASM at any developmental stage and geographical population, and are sensitive enough to be used in quarantine institutions. The demonstration of primer availability is carried out through a series of PCR reactions, followed by gel electrophoresis of the PCR product, and the presence or absence of the target band on the gel is used to determine whether the primer has a specific identification effect. For example, in Figure 6, only the four samples with DNA from ASM as the template were successfully amplified, none of those with DNA templates from other species were amplified. This indicates that the primer has a specific identification effect on ASM. After the products of this series of PCR results were used for gel electrophoresis, we can see that all reactions achieved the experimental objective, i.e. the primers we developed ASMF/ASMR can specifically identify ASM. This shows that we have achieved our experimental objective and  leads us to the general conclusion that the primers (ASMF/ASMR) we developed can specifically identify ASM and can be used in quarantine facilities and ports to prevent the spread of ASM. The principle of molecular identification based on SS-COI is to use a series of PCR products to conduct gel electrophoresis, and to prove whether the primer is available by observing whether there is a band of expected size. If a band of expected size is observed, it indicates that the primer is specific for the target species. Reference can be made to this study: https://doi.org/10.1111/jen.12972.

In addition, as mentioned in introduction, Lymantria apicebrunnea has outbreaked in some parts of China in recent years. It is a little-known species and is very similar to Lymantria dispar in morphology, so that the relevant departments mistook it for Asian spingy month. Based on the morphological similarity between this species and Asian spongy month, we think it is necessary to make a preliminary exploration of its genetic relationship, the phylogenetic and genetic distance analyses were therefore carried out. The results show that Lymantria apicebrunnea is indeed closely genetically related to Asian spongy moth. Therefore, we designed the ASM-specific primers to include L. apicebrunnea as a closely related species, which further ensures the specificity of the ASM-primers. The results also remind us of the need for further control of this pest in the future to prevent it from becoming a major quarantine pest like Asian spongy moth. besides, the results can also provide direction for further research.

Point 9: This section lacks a discussion of the efficiency of the technique and how it differs from previous molecular techniques. A detailed comparison of the results obtained (technically) with those of many previous authors seeking to validate the results is lacking.

Response 9: Thank you so much for your careful check. We supplemented the efficiency of the technique and how it differences from previous molecular techniques: “Many advanced methods have been developed for moth identification, such as DNA barcoding, loop-mediated isothermal amplification (LAMP) technology, and recombinase polymerase amplification (RPA) technology, and the most widely used method for the rapid identification of animals is DNA barcoding. Although DNA barcoding has often been used for the identification of economically important insects, one limitation of this method is that it requires a DNA barcode database with a sufficient number of species to make accurate identifications [61]. Second, LAMP technology and RPA technology have been widely used because of their high sensitivity. However, the high sensitivity and amplification rates of LAMP can increase the rate of false positives. The loop primers used in LAMP increase the amplification efficiency but are vulnerable to generating false positives because of the non-specific pairing of the loop primers [62]. The RPA technique is an effective species identification approach that does not require a thermal cycler, but the partial non-specific amplification and false positives associated with reactions conducted at a constant temperature are major shortcomings that merit consideration. In addition, the TwistAmp® Basic Kit required for RPA is expensive, it is not suitable for large-scale testing in ports and it is not available in some remote locations”(line 444-460).

The supplement of comparing the results obtained (technically) in detail with the results that many previous authors tried to verify, what we want to explain is that no matter which technology is used for rapid identification, the authors want to demonstrate is that their methods can be used for rapid identification of target species. The difference is that different methods or systems will have different efficiency in identifying target species. Therefore, we supplemented the comparison between our method and previous research in the discussion: “Our tests showed that our ASM-specific primers have high sensitivity and specificity and thus would be effective for the rapid detection of ASMs in quarantine ports. Compared to previous studies, the ASM-specific primers that we designed are effective for detecting ASM samples from all geographic populations in China and developmental stages, including from the first instar to the last instar stages, which was not performed in previous studies. Furthermore, the specific primers we designed are effective under a wider range of annealing temperatures (49–56°C) compared with the primers used in previous studies. The results of previous studies may have involved only one suitable annealing temperature. In such a scenario, the replacement of the PCR instrument during detection might lead to detection errors due to temperature differences among PCR instruments. The ability of our primers to function effectively under a wide range of annealing temperatures can eliminate the effects of temperature differences among PCR instruments and enhance the accuracy of detecting the target species. We added different proportions of ASM DNA to samples of DNA of other moths to evaluate the sensitivity of our primers, and the results indicated that the primers could accurately detect ASMs at a ratio as low as 1:100.”(line 481-496).

Point 10: What is the conclusion of the work?

Response 10: Thank you for pointing out this problem in manuscript. We have added the conclusion to the manuscript: “Invasions of ASMs are considered more serious threats to forest resources than invasions of ESMs because of their ability to induce greater amounts of damage. The implementation of quarantine measures is essential to prevent the spread of ASMs. ASMs are native to and widely distributed in China. We extensively sampled ASMs in China and used SS-COI PCR to rapidly and accurately identify ASMs. This method is an improvement to existing methods and could be used in quarantine laboratories. It could thus be useful for monitoring and preventing the spread of ASMs to other regions.” (line 509-515)

Reviewer 2 Report

I found this paper well done from an experimental point of view. The authors have conducted the experiments with a high degree of attention and precision and therefore I believe in their method.

On the other hand, from a conceptual point of view, the paper has to be improved. The English language is poor and often inadequate. The introduction section has to be rewritten, explaining better the complex situation of the Asian Lymantria species.

The discussion section should be shortened. Often the authors report again the results in this section and then discuss them. This makes reading cumbersome and lengthens the discussion for no reason. The results are already listed in the results section, in the discussion they only need to be discussed.

A paragraph in the discussion section concerns the utility of mtDNA and nuclear DNA in taxonomic and phylogenetic studies. I think that this is out of the aims of the paper and strongly suggest deleting it, referring only to DNA barcode as a tool in taxonomic issues.

Detailed comments are in the attached file

Author Response

Dear Reviewer,

On behalf of my co-authors, we thank you very much for giving us an opportunity to revise our manuscript, and we appreciate very much for your positive and constructive comments and suggestions on our manuscript entitled “Improvement of the identification technology for Asian Spongy Moth, Lymantria dispar Linnaeus, 1758 (Lepidoptera: Erebidae) based on SS-COI”. We have studied the comments carefully and have made revisions marked in highlight (The blue highlight is a language improvement that native English speakers help us make. It does not change the original meaning of the sentence, but merely adjusts the grammar and the order of the words to more closely resemble the English expression. The yellow highlighting is our point-to-point revision based on the reviewers' comments). Attached please find the revised version, which we would like to submit for your kind consideration.

We tried our best to improve the manuscript and made some changes in the manuscript. These changes will not affect the content and framework of the paper. We hope that our revision could address all of the questions by the reviewers.

Thank you very much for your considering our manuscript for publication. We are looking forward to hearing from you soon.

Sincerely yours,

Prof. Juan Shi

Beijing Key Laboratory for Forest Pest Control, Beijing Forestry University,Sino-French Eurasian Forest Invasive Biology Laboratory, Beijing Forestry University

P.O. Box 113, 35 Qinghua East Road, Haidian District, Beijing 100083, P. R. China

Tel: +86-10-62336423

Fax: +86-10-62336423

Response to Reviewer 2 Comments

Point 1: On the other hand, from a conceptual point of view, the paper has to be improved. The English language is poor and often inadequate. The introduction section has to be rewritten, explaining better the complex situation of the Asian Lymantria species.

Response 1: We appologize for the language problems of our manuscript and the revised manuscript was further improved and revised with the help of the native English speakers. We really hope that the language level can be substantially improved.

We have rewritten the introduction according to your suggestions, and explained the situation of spongy month more clearly:”Lymantria dispar Linnaeus 1758 (Lepidoptera: Erebidae: Lymantriinae), which is commonly known as spongy moth, is an omnivorous pest native to Europe, Asia, and North Africa that have also been introduced to the North America [11]. Spongy moths have been divided into three subspecies according to their distribution and whether the female is capable of flight. Lymantria dispar dispar Linnaeus 1758 mainly occurs in Europe and North America and is commonly referred to as European spongy moth (ESM). Lymantria dispar asiatica Vnukovskij 1926 and Lymantria dispar japonica Motschulsky 1860, which are distributed in Asia and both are referred to as Asian spongy moth (ASM)”

Point 2: The discussion section should be shortened. Often the authors report again the results in this section and then discuss them. This makes reading cumbersome and lengthens the discussion for no reason. The results are already listed in the results section, in the discussion they only need to be discussed.

Response 2: We gratefully appreciate for your valuable suggestion. As you said, there will be some duplication of results in the discussion, so we removed the description of results from the discussion. Please see the discussion for more details (line 387-508)

Point 3: A paragraph in the discussion section concerns the utility of mtDNA and nuclear DNA in taxonomic and phylogenetic studies. I think that this is out of the aims of the paper and strongly suggest deleting it, referring only to DNA barcode as a tool in taxonomic issues.

Response 3: Thank you for your constructive suggestions which we greatly appreciate. According with your advice, we have revisited the manuscript and removed the discussion on the use of mtDNA and nuclear DNA in phylogenetic analyses. Only the role of DNA barcoding in taxonomy has been retained.

The revised text reads as follows: “Mitochondrial DNA markers that are polymorphic within and among species and populations are also effective for species identification [53,54].

The COI gene has been shown to be effective for the identification of various animal species. In 2003, Hebert first proposed that the COI gene could be used for species identification. Since then, DNA barcoding technology has received wide research attention [55-57]. DNA barcode technology is now widely used for species identification and in studies of molecular evolution, genetic variation, and biodiversity. Some of the reasons for its wide use include its low cost, ease of implementation, and robustness to interference from the external environment. DNA barcoding with COI genes has been shown to be effective for the identification of various animal species [58]. In the animal kingdom, more than 95% of the species can be accurately identified at the species level using this approach [59]. Kang used DNA barcodes to detect Lymantria species in Korea and showed that this approach can distinguish between closely related species [60] (line 422-435).

We have also made the following modifications according to your attached file:

Point 4Line 88: Iow temperatures to hatch compared to eggs of the EGM

Response 4: We are sorry for the writing problems. We have made changes in the new manuscript:Eggs of the Asian spongy moth strain required less exposure to low temperatures to hatch compared to eggs of the ESM (line 92).

Point 5:LINE 100: what do you mean with "etc."? You should cite your work or, alternatively, explain what you did.

Response 5: We are very sorry for our negligence in reference, and thank you for your concern. We have supplemented the references in the new manuscript: ”We have been working on a comprehensive study of the biology, genetic variation of spongy moth as an economically important pest [24,28-35]” (line 101-103).

Point 6: Move Lepidoptera: Erebidae the line above, after Lymantria species. I imagine that all species belonging to Lymatria genus are Erebidae and not only L. apicebrunnea.

Response 6: Thank you for pointing this out in the manuscript. We have revised it in the new manuscript: ”Several morphologically similar species in the genus Lymantria (Lepidoptera: Erebidae) in China can be confused with spongy moths such as Lymantria apicebrunnea Gaede, 1932, Lymantria similis Moore 1879, Lymantria fumida Butler 1877, Lymantria xylina Swinhoe 1903, Lymantria mathura Linnaeus 1758.”

Point 7: Line125: what is mu? Line 128: what is RMB?

Response 7: We are sorry that we did not convert the unit. We have revised in the new manuscript: According to the statistics, only in one small city, the area of the outbreak reached 8.34 km2 (line 129). A total of 2.86 million CNY have been invested to mitigate the deleterious effects of this pest in its original range (line 131).  

Point 8: Line 128: please substitue with "of greatest importance"

Response 8: We gratefully appreciate for your valuable suggestion. We have replaced with “of greatest importance“ (line 140).

Point 9: Line 201: Please change "chest" in torax and "foot" in leg.

Response 9: We apologise for our writing errors and thank you very much for your suggestions. We have made corrections in the new manuscript: “for adult moths, thorax or leg tissue was obtained. Leg tissue is often recommended for extracting genomic DNA from various types of insects to maintain the integrity of specimens.” (line 199 ).

Point 10: Line 222: I don't understand what you mean. Did you find introns? Please, explain.

Response 10: We apologise for not being clear here, what we are trying to convey is that when performing phylogenetic analysis, multiple sequence alignment is first performed and in the process, all gaps in the aligned sequences are removed. The aligned and clipped sequences are then used for phylogenetic analysis. To avoid ambiguity, we have removed this sentence from the manuscript (line 221).

Point 11: Line 321: which species? Please, specify again.

Response 11: Thank you so much for your careful check. We have made refinements in the revised manuscript: “PCR and agarose gel electrophoresis were conducted using the DNA of the four species (L. dispar, L. xylina, L. monacha, and L. apicebrunnea) as templates.” (line 319).

Point 12: Line 345 "negative control". What do you mean? Deionized sterilized water? Control mix (i.e., the PCR mix without DNA)? You had to use the control mix. This is not clear from the text and therefore, please explain and correct it throughout the paper.

Response 12: We are very grateful for your careful review and suggestions. In the manuscript, negative control refers to PCR mix without DNA, and we have made detailed corrections and clarifications in the manuscript.

Point 13: Line 410: substitute "impeding" with "preventing"

Response 13: We gratefully appreciate for your valuable suggestion. We have changed “impeding” to “preventing” (line 408).

Reviewer 3 Report

The submitted MS dedicated to development targeting PCR primers for molecular identification of certain lineages within the spongy moth, Lymantria dispar sensu lato, one of the most important quarantine pest among Lepidoptera order and Insects as a whole.

In general, them MS is well-written, and definitely should be published in “Insects”, however it consist a number of minor errors and few general moments, that must be corrected prior acceptance.

Minor comments (see attached pdf file)

General comments.

 (1) Images (at least in the uploaded pdf file) are of a poor quality. Depicted values (see, for instance fig. 4 and 5) are hardly seen. In all cases, the images are of a half width of journal page. I would recommend increase size to full page for figures 1,2,4.5.  Also, why not to put two gel images in a row, to reduce number of figures and save some space?

(2)  If mentioned for the first time, all scientific names of the generic and specific rank should be followed by the author's name (authors' names) and year of publication. It is common practice for taxonomical journals and I would recommend to follow it in the MS.

(3) Section 3.2. Phylogenetic analysis of Lymantria of the “Results” chapter.

I would recommend re-phrase this section. I do understand what authors are talking about, but phrases like “It is well know L. postalba and L. albescens were the two most closely related species (genetic distance of 0)” somewhat confusing. If there is no genetic divergence, why should we consider them as a separate species? It is necessary here and through the entire 3.2. section to point out, that “genetic distance of 0 (as well as all genetic distances in question) referred ONLY to  studied COI fragment.   

Author Response

Dear Reviewer,

On behalf of my co-authors, we thank you very much for giving us an opportunity to revise our manuscript, and we appreciate very much for your positive and constructive comments and suggestions on our manuscript entitled “Improvement of the identification technology for Asian Spongy Moth, Lymantria dispar Linnaeus, 1758 (Lepidoptera: Erebidae) based on SS-COI”. We have studied the comments carefully and have made revisions marked in highlight (The blue highlighting is a linguistic enhancement that we have invited native English speakers to help us with. It does not change the original meaning of the sentence, but merely adjusts the grammar and the order of the words to more closely resemble the English expression. The yellow highlighting is our point-to-point revision based on the reviewers' comments). Attached please find the revised version, which we would like to submit for your kind consideration.

We tried our best to improve the manuscript and made some changes in the manuscript. These changes will not affect the content and framework of the paper. We hope that our revision could address all of the questions by the reviewers.

Thank you very much for your considering our manuscript for publication. We are looking forward to hearing from you soon.

Sincerely yours,

Prof. Juan Shi

Beijing Key Laboratory for Forest Pest Control, Beijing Forestry University,Sino-French Eurasian Forest Invasive Biology Laboratory, Beijing Forestry University

P.O. Box 113, 35 Qinghua East Road, Haidian District, Beijing 100083, P. R. China

Tel: +86-10-62336423

Fax: +86-10-62336423

Response to Reviewer 3 Comments

Point 1: Images (at least in the uploaded pdf file) are of a poor quality. Depicted values (see, for instance fig. 4 and 5) are hardly seen. In all cases, the images are of a half width of journal page. I would recommend increase size to full page for figures 1,2,4.5.  Also, why not to put two gel images in a row, to reduce number of figures and save some space?

Response 1: We are very sorry that the quality of the pictures has affected the reading of the manuscript. Thank you for your kind suggestions. The quality of the images is very valuable for improving the accuracy of the manuscript. We have uploaded the high-resolution image again, and enlarged the size of the image to the whole page according to your suggestions, so that it can be displayed more clearly. In addition, we have moved and modified the position of gel: ① Move the gel at the bottom of Figure 7 to the upper right corner to save space (line 338). ② The two gel images, Fig. 10 and Fig. 11, have been combined in one row because they are both the result of sensitivity testing for specific primers, so that combining them does not change the original content of the manuscript (line 369). This will also reduce the number of images. However, because other gel images represent different detection purposes (for example, Figure 6 is the result of specificity test on ASM specific primers, and Figure 7 is the result of exploring the impact of different annealing temperatures on primers), putting two gel images in a row may cause readers' misunderstanding, so we think that not merging other gel images may be beneficial to readers' understanding.

Point 2:  If mentioned for the first time, all scientific names of the generic and specific rank should be followed by the author's name (authors' names) and year of publication. It is common practice for taxonomical journals and I would recommend to follow it in the MS.

Response 2: Thank you very much for the positive comments and constructive suggestions. We have rechecked the manuscript and added the author and year after the scientific name mentioned for the first time: Lymantria dispar Linnaeus, 1758 (line 3); Lymantria Hübner, 1819 (line 22); Lymantria dispar dispar Linnaeus 1758 (line 70); Lymantria dispar asiatica Vnukovskij 1926 (line 72); Lymantria dispar japonica Motschulsky 1860 (72); Lymantria apicebrunnea Gaede 1932 (line 118); Lymantria similis Moore, 1879 (line 119); Lymantria fumida Butler, 1877 (line 119); Lymantria xylina Swinhoe, 1903 (line 120); Lymantria mathura Linnaeus, 1758 (line 120); Lymantria mathura Moore, 1865 (line 176); Lymantria nebulosa Wileman, 1910 (line 176);  Lymantria marginata Walker, 1855 (line 177).

Point 3: Section 3.2. Phylogenetic analysis of Lymantria of the “Results” chapter.

I would recommend re-phrase this section. I do understand what authors are talking about, but phrases like “It is well know L. postalba and L. albescens were the two most closely related species (genetic distance of 0)” somewhat confusing. If there is no genetic divergence, why should we consider them as a separate species? It is necessary here and through the entire 3.2. section to point out, that “genetic distance of 0 (as well as all genetic distances in question) referred ONLY to  studied COI fragment.   

Response 3: Thank you for pointing out this problem in manuscript. We refer to your suggestion and point out in the manuscript that all genetic distances are calculated based on COI fragments. And we have rewritten this paragraph. We objectively stated the results of phylogenetic analysis in the results section as follows: The COI barcode fragments amplified from the collected samples and the sequences of other Lymantria species downloaded from the NCBI database were used to build phylogenetic trees (Figure 4) and genetic distance matrices (Figure 5). It can be seen that L. apicebrunnea and Lymantria schaeferi (Schintlmeister, 2004) are clustered together and their genetic distance based on the COI barcode fragment is 0. This finding is consistent with the results of the BLAST comparison, which indicated that these two species are closely related. The genetic distance between the two ASM subspecies, Lymantria dispar asiatica and Lymantria dispar japonica, based on our selected fragment was 0. The genetic distance between the two species Lymantria albescens (Hori and Umeno, 1930) and Lymantria postalba (Inoue, 1956) in Japan was also 0. The species most closely related to L. dispar according to our data was Lymantria umbrosa (genetic distance of 0.02 to Lymantria dispar). The genetic distances of several species in our analysis, including L. schaeferi, L. xylina, and L. apicebrunnea, to L. dispar ranged from 0.06 to 0.07 ”(line 289-301).

With regard to your question "If there is no genetic divergence, why should we consider them as a separate species?" , given the structure of the manuscript, we put it in the discussion. This is due to the fact that there is still some controversy about the taxonomic status of L. albescens and L. postalba, with some scholars arguing that they should be the same species (Djoumad, 2020), and our results also show little genetic divergence based on at least the COI fragment. A detailed discussion of this revised section is as follows: Our findings revealed that L. postalba and L. albescens are closely related, with a genetic distance between them based on COI fragments of 0. The taxonomic status of the two species remains controversial, with some re-searchers arguing that "L. albescens and L. postalba are at best two forms of a single species"[63]. Similarly, we found that the genetic distance between L. apicebrunnea and L. schaeferi based on COI fragments was also 0. This discovery has raised questions regarding the taxonomic status of these two species. As we did not have specimens of L. schaeferi and our phylogenies were constructed using only COI fragments, additional research is needed to re-evaluate the taxonomic status of these two species. What is clear from our data is that L. apicebrunnea and L. dispar are closely related. Coupled with their high morphological similarity and ability to feed on the same hosts, means that special efforts should be made to prevent the spread of L. apicebrunnea so that it does not become a major pest.”(line 471-480)

Reviewer 4 Report

Dear Authors,

I read your manuscript on improved identification technique for Asian spongy moth. Your manuscript is well written and represents an important solution of the monitoring method to identify the ASM.

Minor English editing changes are needed. Also I made a list of some minor corrections or sugesstions to improve the text.

LINE 21: replace »even« for »always«

LINE 26: »genetic variation varies« - change this into more appropriate phrase

LINE 28: Correct the sentence so it would read:…which can be completed within 2-3 hours using as little as 30 pg…

LINE 29: instead of »update«, »monitor« may be the better expression here

LINES 31-32: It is not clear whether spongy moth and Asian spongy moth are the same or different species or subspecies. Reform this part so that it becomes clearer.

LINE 38: replace word »from« for »based on the«

LINE 40: what is single-headed ASM?

LINES 163-178: This partmust be moved into methods. Instead write a short statement on what was the purpose of your study and what important issues does your method solves.

Figures 4 and 5 need to be enlargedand resolution improved since it is impossible to read the words and numbers. Also the colour legend is missing for Figure 4.

LINES 468-471: I disagree with your statement. For quarantine identification purposes as well as in routine molecular monitoring it is necessary that te person has a proper background and extensive training in order to be aware of all the mistakes that may be done during the procedure. Mixing of the samples must be prevented as well as possibility of pipeting errors. Also the sterility during the procedure is essential to prevent cross contamination of the samples.

Author Response

Dear Reviewer,

On behalf of my co-authors, we thank you very much for giving us an opportunity to revise our manuscript, and we appreciate very much for your positive and constructive comments and suggestions on our manuscript entitled “Improvement of the identification technology for Asian Spongy Moth, Lymantria dispar Linnaeus, 1758 (Lepidoptera: Erebidae) based on SS-COI”. We have studied the comments carefully and have made revisions marked in highlight (The blue highlight is a language improvement that native English speakers help us make. It does not change the original meaning of the sentence, but merely adjusts the grammar and the order of the words to more closely resemble the English expression. The yellow highlighting is our point-to-point revision based on the reviewers' comments). Attached please find the revised version, which we would like to submit for your kind consideration.

We tried our best to improve the manuscript and made some changes in the manuscript. These changes will not affect the content and framework of the paper. We hope that our revision could address all of the questions by the reviewers.

Thank you very much for your considering our manuscript for publication. We are looking forward to hearing from you soon.

Sincerely yours,

Prof. Juan Shi

Beijing Key Laboratory for Forest Pest Control, Beijing Forestry University,Sino-French Eurasian Forest Invasive Biology Laboratory, Beijing Forestry University.

P.O. Box 113, 35 Qinghua East Road, Haidian District, Beijing 100083, P. R. China

Tel: +86-10-62336423

Fax: +86-10-62336423

Response to Reviewer 4 Comments

Point 1: LINE 21: replace »even« for »always«

Response 1: We gratefully appreciate for your valuable suggestion. We have changed “even” to “ always”. (line 23)

Point 2: LINE 26: »genetic variation varies« - change this into more appropriate phrase

Response 2: Thank you for pointing out this problem in manuscript. We have changed “genetic variation varies” to “ high genetic variation”. The new sentence reads as follows: Moreover, sampling has only been conducted in a few locations in China, given that high genetic variation has been detected in ASMs from different regions in China in recent years, the current methods used for identifying ASMs are not sufficiently robust. (line 30)

Point 3: LINE 28: Correct the sentence so it would read:…which can be completed within 2-3 hours using as little as 30 pg…

Response 3: We gratefully appreciate for your valuable suggestion. We have revised it as follows: This improved approach permits identifications of ASMs to be made in 2-3 hours using as little as 30 pg of genomic DNA. (line 32)

Point 4: LINE 29: instead of »update«, »monitor« may be the better expression here

Response 4: We gratefully appreciate for your valuable suggestion. We have changed “update” to “monitor” (line 33)

Point 5: LINES 31-32:  It is not clear whether spongy moth and Asian spongy moth are the same or different species or subspecies. Reform this part so that it becomes clearer.

Response 5: Thank you for pointing out this problem in manuscript. We have made a detailed description about 'spongy month "and" Asian spongy month“. The details as follows: Lymantria dispar (Linnaeus, 1758), which is commonly known as spongy moth, with two subspecies occur in Asia: Lymantria dispar asiatica and Lymantria dispar japonica, collectively referred to as the Asian spongy moth (ASM). The subspecies Lymantria dispar dispar occurs in Europe and is commonly known as the European spongy moth (ESM). (line 34-37)

Point 6: LINE 38: replace word »from« for »based on the«

Response 6: We gratefully appreciate for your valuable suggestion, we have adjusted this sentence as follows: ”which were designed based on cytochrome oxidase I sequences from samples obtained from all sites where ASMs have been documented to occur in China.” (line 44-46)

Point 7: LINE 40: what is single-headed ASM?

Response 7: We are very sorry for our incorrect writing. The correct writing should be "single ASM", and we have corrected this problem in the manuscript: “We show that these primers are effective for identifying single ASM at all life stages and from all ASM populations in China” (line 46)

Point 8: LINES 163-178: This partmust be moved into methods. Instead write a short statement on what was the purpose of your study and what important issues does your method solves.

Response 8: We gratefully appreciate for your valuable suggestion. Considering that the experimental operation steps have been described in detail in the method part, in order to avoid repetition, we have deleted the description of this part of the experimental operation, and briefly introduced the purpose of our experiment and the key problems to be solved. The revised version is as follows: “Here, we improve the method for the rapid and accurate identification of ASMs using a pair of SS-COI primers. A pair of specific primers and a molecular detection system have been established based on the COI gene. This method is simple to operate and takes only 2-3 hours to complete. This improved method will aid the identification of ASMs in China and thus help prevent the spread of ASM to other regions.” (line 166-170)

Point 9: Figures 4 and 5 need to be enlargedand resolution improved since it is impossible to read the words and numbers. Also the colour legend is missing for Figure 4.

Response 9: Thank you for pointing out this problem in manuscript. We have replaced Figure 4 and Figure 5 with high-resolution images and enlarged the size to clearly show the numbers (line 310, 312). As shown in the figure, the color in Figure 4 is used to highlight the species we are concerned about and facilitate observation and analysis. Unlike the color legend in Figure 5, which represents the range of values, Figure 4 can be clearly displayed without the color legend. For this reason, we did not add a color legend to Figure 4.

Point 10:  LINES 468-471: I disagree with your statement. For quarantine identification purposes as well as in routine molecular monitoring it is necessary that te person has a proper background and extensive training in order to be aware of all the mistakes that may be done during the procedure. Mixing of the samples must be prevented as well as possibility of pipeting errors. Also the sterility during the procedure is essential to prevent cross contamination of the samples.

Response 10: Thank you for pointing out this problem in manuscript and we apologise for the lack of rigour of expression here. As you point out, molecular testing is an important tool for quarantine and identification of pests, and operators must have rigorous molecular training and be careful to ensure the accuracy of identification. On the other hand, the sterility of the testing environment is another key element for successful identification. The original descriptions in our manuscript are somewhat one-sided. To prevent ambiguity, we have removed these descriptions from the manuscript and instead compared the advantages and disadvantages of the different molecular assays in an objective manner. The revised contents are as follows: “Many advanced methods have been developed for moth identification, such as DNA barcoding, loop-mediated isothermal amplification (LAMP) technology, and recombinase polymerase amplification (RPA) technology, and the most widely used method for the rapid identification of animals is DNA barcoding. Although DNA barcoding has often been used for the identification of economically important insects, one limitation of this method is that it requires a DNA barcode database with a sufficient number of species to make accurate identifications [61]. Second, LAMP technology and RPA technology have been widely used because of their high sensitivity. However, the high sensitivity and amplification rates of LAMP can increase the rate of false positives. The loop primers used in LAMP increase the amplification efficiency but are vulnerable to generating false positives because of the non-specific pairing of the loop primers [62]. The RPA technique is an effective species identification approach that does not require a thermal cycler, but the partial non-specific amplification and false positives associated with reactions conducted at a constant temperature are major shortcomings that merit consideration. In addition, the TwistAmp® Basic Kit required for RPA is expensive, it is not suitable for large-scale testing in ports and it is not available in some remote locations [41]. In light of these considerations, we used SS-COI PCR because our aim was to explore the utility of an approach that could be used in real-world quarantine situations. We showed that our new primers were highly sensitive and specific and thus that they could be used for large-scale testing in ports. We believe that the SS-COI PCR is a cost-effective approach with high specificity and sensitivity that would be a convenient tool for quarantine agencies that do not have easy access to well-equipped laboratories.” (line 445-461).

Round 2

Reviewer 1 Report

All comments made were answered correctly

Reviewer 2 Report

Dear authors,

The paper has been greatly improved and is now publishable.